# Prebiotic Galactooligosaccharide Supplementation in Adults with Ulcerative Colitis: Exploring the Impact on Peripheral Blood Gene Expression, Gut Microbiota, and Clinical Symptoms

**DOI:** 10.3390/nu13103598

**Published:** 2021-10-14

**Authors:** Bridgette Wilson, Özge Eyice, Ioannis Koumoutsos, Miranda C. Lomer, Peter M. Irving, James O. Lindsay, Kevin Whelan

**Affiliations:** 1Department of Nutritional Sciences, King’s College London, London SE1 9NH, UK; bridgette.wilson@kcl.ac.uk (B.W.); miranda.lomer@kcl.ac.uk (M.C.L.) peter.irving@gstt.nhs.uk (P.M.I.); 2Department of Nutrition and Dietetics, Guys and St Thomas’ NHS Foundation Trust, London SE1 7EH, UK; 3School of Biological and Chemical Sciences, Queen Mary University of London, London E1 4NS, UK; o.eyice@qmul.ac.uk; 4Department of Gastroenterology, Guys and St Thomas’ NHS Foundation Trust, London SE1 7EH, UK; ioannis.koumoutsos@southend.nhs.uk; 5Department of Gastroenterology, Barts Health NHS Trust, London E1 1FR, UK; james.lindsay8@nhs.net; 6Blizard Institute, Barts and the London School of Medicine and Dentistry, London E1 2AT, UK

**Keywords:** prebiotics, ulcerative colitis, gene expression, microbiota, microbiome

## Abstract

Prebiotics may promote immune homeostasis and reduce sub-clinical inflammation in humans. This study investigated the effect of prebiotic galactooligosaccharide (GOS) supplementation in colonic inflammation. Seventeen patients with active ulcerative colitis (UC) consumed 2.8 g/d GOS for 6 weeks. At baseline and 6 weeks, gene expression (microarray), fecal calprotectin (ELISA), microbiota (16S rRNA), short-chain fatty acids (SCFAs; gas-liquid chromatography), and clinical outcomes (simple clinical colitis activity index (SCCAI), gastrointestinal symptom rating scale (GSRS), and Bristol stool form scale (BSFS)) were measured. Following prebiotics, clinical scores (SCCAI), fecal calprotectin, SCFAs, and pH were unchanged. Five genes were upregulated and two downregulated. Normal stool proportion (BSFS) increased (49% vs. 70%, *p* = 0.024), and the incidence (46% vs. 23%, *p* = 0.016) and severity (0.7 vs. 0.5, *p* = 0.048) of loose stool (GSRS), along with urgency (SCCAI) scores (1.0 vs. 0.5, *p* = 0.011), were reduced. In patients with a baseline SCCAI ≤2, prebiotics increased the relative abundance of *Bifidobacterium* from 1.65% (1.97) to 3.99% (5.37) (*p* = 0.046) and *Christensenellaceae* from 0.13% (0.33) to 0.31% (0.76) (*p* = 0.043). Prebiotics did not lower clinical scores or inflammation but normalized stools. *Bifidobacterium* and *Christensenellaceae* proportions only increased in patients with less active diseases, indicating that the prebiotic effect may depend on disease activity. A controlled study is required to validate these observations.

## 1. Introduction

Prebiotics are substrates that are selectively utilized by host microorganisms to confer a health benefit [1]. Mechanisms of health benefit include augmentation of bifidobacteria and other bacterial genera, increased immunoregulatory short-chain fatty acid (SCFA) production, and direct and indirect modulation of inflammation [2,3]. The two most extensively researched prebiotics are inulin-type fructans (ITF) and galactooligosaccharides (GOS).

Prebiotics increase bifidobacteria in vitro and in vivo [4,5]. Bifidobacteria modulate pH through the production of lactate and acetate, which contributes to butyrate production via acetate cross-feeding; compete with pathogens for resources; and may favorably modulate immune signaling [5]. Further, a lower concentration of *Bifidobacterium* species is associated with fewer interleukin-10-secreting dendritic cells in inflammatory bowel disease (IBD) [6]. Lower bifidobacteria concentrations have been described in gastrointestinal inflammation as seen in inflammatory bowel disease (IBD) and in sub-clinical inflammation as described in irritable bowel syndrome (IBS) [7,8]. Although observational, the association between low bifidobacteria and inflammation supports the theory that prebiotics may reduce inflammation through augmenting bifidobacteria.

There are two main types of IBD, ulcerative colitis (UC) and Crohn’s disease; both are chronically relapsing and remitting inflammatory diseases of the bowel with both genetic and environmental causes. While UC is characterized by continuous superficial inflammation isolated in the colon, Crohn’s disease is characterized by patchy, penetrating inflammation that can be located anywhere along the length of the gastrointestinal tract. Inflammation and symptoms of IBD are managed through medication, although there is an increasing interest in dietary management.

In IBD, inflammation is driven by an immune response to commensal gastrointestinal microbiota, which activates toll-like receptor (TLR) signaling [9]. Altered gastrointestinal microbiota, a reduced gastrointestinal barrier function, and increased pro-inflammatory signals from bacteria encroaching on the gut barrier are implicated in the inflammation seen in IBD [10].

Clinical trials of prebiotics in IBD have largely focused on inulin-type fructans and have shown mixed results. In active UC, 12 g/d ITF (oligofructose) reduced fecal calprotectin compared to baseline after 2 weeks whereas placebo did not [11]. However, in active Crohn’s disease, 15 g/d of ITF (oligofructose/inulin) worsened abdominal pain and did not increase clinical response or remission rates compared to placebo, although it reduced IL-6-positive dendritic cells and increased IL-10-positive dendritic cells in the lamina propria [12]. Further, a recent systematic review and meta-analysis found that modification of the microbiota using prebiotics, probiotics, or synbiotics for 2–8 weeks reduces disease activity in UC but not in Crohn’s disease [13]. These studies suggest that prebiotics may impact gastrointestinal immune signaling but may be more effective and better tolerated in UC.

The degree of polymerization, dose, and type (ITF or GOS) of prebiotic may impact the effect on both functional gastrointestinal symptoms and inflammatory and immune signaling [14,15,16]. Short-chain ITF prebiotics (degree of polymerization: 2–10), but not long-chain (degree of polymerization: 10–60), protect GI cell barrier function in vitro [16], and short-chain ITF and GOS modulate peripheral immune signaling towards homeostasis in vivo [3,17]. In vitro studies have demonstrated that *Bifidobacterium bifidus* are protective against TNFα-induced loss of trans-epithelial electro-resistance [18], of relevance in IBD as anti-TNFα medication is an established therapy in IBD. In addition, GOS prebiotics alter gene expression in intestinal goblet cells in vivo to increase expression of four potentially protective mucosal layer associated proteins (MUC2, TFF3, RETNLB, and CHST5) [19]. Therefore, prebiotics provide an intriguing strategy for dietary research in gastrointestinal inflammation. However, the effect of prebiotics on immune-related gene expression in mild colonic inflammation is unknown, and the use of lower doses of GOS and their use in UC at a dose that does not induce functional symptoms needs clarifying. The aim of this study was to investigate the impact of prebiotic GOS on gene expression, inflammation, disease parameters, and microbiota in mildly active UC.

## 2. Materials and Methods

This was an open-label study that aimed to investigate the effect of 6 weeks of GOS supplementation (2.8 g/d active GOS (Bimuno GOS, Clasado Biosciences, Reading, UK)) in patients with mildly active UC.

### 2.1. Participant Selection

Patients were recruited from gastroenterology clinics at Guy’s and St Thomas’ NHS Foundation Trust and Barts Health NHS Trust (London, UK). As this was an exploratory study to inform hypotheses for future research, a pragmatic sample size of 18 patients was selected.

The inclusion criteria were: adults (16–65 years); active UC (judged by gastroenterologist opinion and supported by at least one of the following objective measures of disease activity in the previous two months: (i) CRP above normal for referring center; (ii) fecal calprotectin greater than 150 µg/g (identified as a marker of active disease) [20]; or (iii) endoscopic evidence of active disease recorded in clinical notes); absence of other major medical conditions (e.g., diabetes, major psychiatric disorder, current eating disorders, major food allergy), gastrointestinal disorders (e.g., coeliac disease), known enteropathogens, or previous GI surgery (except cholecystectomy, appendectomy, and hemorrhoidectomy); no bowel preparation or microbiota-modifying treatment (e.g., antibiotics, prebiotics, probiotics) within 4 weeks of trial commencement; not pregnant or breastfeeding.

Exclusion criteria were: commencement of azathioprine, mercaptopurine, or thioguanine therapy within the preceding 12 weeks or altered dose in the previous 6 weeks; commencement of 5-ASA (mesalazine) less than 8 weeks previously or altered dose in the previous 2 weeks; current use of medication steroids, non-steroidal anti-inflammatory drugs, methotrexate, ciclosporin, tacrolimus, or biologic drugs, e.g., infliximab, vedolizumab, golimumab, adalimumab; regular use of supplements/medications that could affect the luminal gastrointestinal environment (e.g., orlistat, lactulose).

### 2.2. Trial Protocol

The trial consisted of a one-week baseline data collection period followed by a six-week open-label prebiotic supplementation period. Patients meeting the inclusion criteria were asked to maintain their usual diet, medication, and smoking habits and were asked not to consume any products containing prebiotics or probiotics during the trial. At baseline and 6 weeks, anthropometry (weight, height, BMI) was recorded, a blood sample was drawn for microarray analysis, and a fresh stool sample was collected to measure fecal calprotectin, microbiota, SCFAs, and pH. Patients were contacted by telephone after one, three, and five weeks of the intervention to assess adherence and record any adverse events or changes to medication. Telephone reviews were used to enquire about adverse health events throughout the trial. Patients who breached the protocol during the trial were withdrawn. Patients wishing to discontinue the trial for any reason were free to withdraw.

During the trial, patients consumed 2.8 g/d of GOS by mixing it with 300 mL water and drinking it once daily for six weeks. The GOS was provided in 2.8 g sachets that contained doses of the active ingredient equivalent to the dose used in a previous prebiotic and immunity study in elderly patients [17]. Adherence was measured by direct questioning at each telephone review and by counting the number of unused sachets left at the end of the study. For inclusion in the per protocol population, participants had to report satisfactory adherence (≥80% sachets consumed as assessed by telephone review) and provide blood and stool samples for analysis.

### 2.3. Outcome Measures

The primary outcome was to identify changes in expression of any immune-related gene between baseline and prebiotics using a microarray of all genes expressed in the peripheral blood. Secondary outcomes included changes from baseline in stool microbiota (16S rRNA sequencing), markers of inflammation (calprotectin ELISA) and fermentation (SCFAs and pH), disease activity (SCCAI [21]), gastrointestinal symptoms (gastrointestinal symptom rating scale (GSRS) [22]), and stool form and frequency (Bristol stool form scale (BSFS) [23] and bowel diary).

### 2.4. Gene Expression Analysis in Blood

A non-fasting, venous blood sample was collected at baseline and 6 weeks using standard venipuncture. A 2 mL sample was collected in PAXgene blood tubes containing 6.9 mL of RNA stabilizing solution (PreAnalytiX). These were stored at room temperature for two hours to allow complete cell lysis before being stored at −80 °C until microarray.

The microarray was performed by Eurofins (Denmark) using an Affymetrix HTA 2.0 to quantify all coding and noncoding human genes in blood. Briefly, RNA was isolated and cDNA synthesized before labelling and hybridization using the HTA 2.0 Affymetrix microarray gene library (NetAffx). Gene expression was quantified using a robust multichip average with signal space transformation, an automated process within the Transcriptome Analysis Console (TAC version 4.0, ThermoFisher, Waltham, MA, USA), and expressed as the log_2_ of Tukey’s bi-weight average signal intensity to correct for outliers. Fold-change was calculated and expressed as the log_2_ fold-change between baseline and 6 weeks.

Data were compared to assess differential gene expression between baseline and 6 weeks. Genes with ≥1.5 log_2_ fold-change in expression were identified using TAC software (version 4.0); a threshold of ≥1.5 fold difference in expression was selected as this was stringent enough to remove background noise without missing biologically relevant differences in gene expression [24]. Differentially expressed genes were identified using BLAST (www.ncbi.nlm.nih.gov/BLAST, accessed on 30 March 2018) to find the human gene sequence and the location (chromosome, strand, start and stop codons) within the human genome using the Ensemble public gene database (www.ensemble.org, accessed on 30 March 2018).

### 2.5. Fecal Analysis

A whole fresh stool sample was collected, placed on ice, and processed within one hour of passage by homogenization in a stomacher for 4 min before aliquots were stored at −80 °C for later analysis of calprotectin, SCFAs, and microbiota.

### 2.6. Calprotectin Enzyme-Linked Immuno-Sorbent Assay

Calprotectin was extracted from stool using a kit and standard protocol (Calpro AS, Norway). An enzyme-linked immuno-sorbent assay (ELISA) (FireFly Scientific CAL0100 Calprotectin Test Kit) was performed according to the manufacturer’s protocol. Calprotectin concentration was read on a microplate reader at an optical density of 405 nm. A logarithmic curve was created from the kit standards and calprotectin was calculated for each sample, adjusted for dilution (ng/mL × 2500), and converted to µg/g for comparison. All reagents were supplied with the kit.

### 2.7. Fecal Short-Chain Fatty Acids and pH

Fecal SCFAs were extracted using buffer (0.1% mercury, 1% phosphoric acid with 0.0045% 2,2-dimethylbutyric acid internal standard (Sigma, UK)) and quantified using gas liquid chromatography performed on a 7890A gas chromatograph (Agilent Technologies, Santa Clara, CA, USA). The oven was programmed with an initial temperature of 80 °C and increased by 10 °C/min up to 145 °C and then 100 °C/min up to 200 °C to complete elution. Concentrations of the six SCFAs were determined for each sample in duplicate using the equation from the linear regression curve (area: concentration) for each SCFA calibration curve from the Agilent Chromatogram database (Agilent Technologies, Santa Clara, CA, USA) [25]. To correct for variations in stool water content, SCFA concentrations were calculated per gram of dry weight. Dry weight was calculated by drying a known weight of stool at ~100 °C for 24 h or until constant weight was achieved within 0.01 g.

Fecal pH was measured directly on fresh stool using a calibrated pH probe (InLab^®^ Solids Pro, Mettler Toledo, Greifensee, Switzerland) and meter (FE20 Benchtop pH meter, Mettler Toledo, Greifensee, Switzerland).

### 2.8. Microbiota Sequencing

Total genomic DNA was extracted from 500 mg of fecal samples using a QIAamp Fast DNA Stool Mini kit (Qiagen, UK). Bacterial and archaeal 16S rRNA genes were amplified by PCR using 16S rRNA gene-specific primers (515F-806R [26]) and a Mastercycler Pro thermal cycler (Eppendorf UK Ltd., Stevenage, UK) with MyTaq Red DNA Polymerase (Bioline Reagents Ltd., London, UK). Amplification conditions were as follows: initial denaturation at 95 °C for 5 min; 35 cycles of 95 °C for 1 min, 55 °C for 1 min, and 72 °C for 1.5 min; and a final elongation step at 72 °C for 5 min.

PCR products were cleaned using a Charge Switch PCR Clean-up Kit (Invitrogen Life Technologies, Waltham, MA, USA), quantified with a Qubit dsDNA BR Assay Kit using a Qubit 2.0 Fluorometer (Invitrogen, CA, USA), and prepared for sequencing as described by Caporaso et al. (2012). Sequencing was performed using an Illumina MiSeq platform (300 bp paired-end, Illumina, Inc, San Diego, CA, USA). Merging of the sequences and operational taxonomic unit (OTU) picking were carried out by USEARCH v8 [27] at 97% similarity cut-off. Chimeras were removed and taxonomy assignments were determined against the Greengenes database [28] using RDP Classifier 2.2 [29] via QIIME software, version 1.6.0 [30]. Sequence datasets were submitted to the National Centre for Biotechnology Information (NCBI) Read Archive under the bioproject accession number PRJNA596163. Both species richness and diversity (Chao1 and Shannon) were calculated using OTU numbers and species abundance data. Microbiota responses to the intervention were compared between active and inactive patients according to SCCAI cut-offs.

### 2.9. Clinical Outcomes

Disease activity was measured using the SCCAI at baseline and 6 weeks; a higher score represented more severe symptoms, and the scale consisted of six main disease features: bowel frequency during daytime (score 0–3); bowel frequency during the night (score 0–2); urgency of defecation (score 0–3); blood in stool (score 0–3); and general wellbeing (score 0–4) and extra-intestinal features (score 1 point each) [21]. A score of ≤2 was considered clinical remission [31]. SCCAI was chosen as a disease activity outcome measure as it has proven validity when compared with more complex disease activity scoring indices but allows assessment without subjecting the participant to further laboratory testing or invasive procedures [21].

A standardized seven-day diary was completed prospectively prior to trial entry and during week 6 to record gastrointestinal symptoms (GSRS), stool frequency and consistency (BSFS), and dietary intake, with energy and nutrient intake calculated using Nutritics^®^.

### 2.10. Statistical Analysis

Clinical data were analyzed in the context of intention-to-treat (ITT) analysis, with data missing due to withdrawal being imputed using the last value carried forward. Where data were missing at random (i.e., patient omission of questionnaire responses), efforts were made to obtain the information from the patient; otherwise, the last value was carried forward. Data were also analyzed for the clinical per protocol population (those completing the study) and the laboratory per protocol population (those completing the study and providing data/samples at both time points).

Continuous data were compared to baseline using paired *t*-tests or Wilcoxon signed-rank tests, as appropriate, and dichotomous outcomes were compared using McNemar analysis. Data for gene expression were compared between baseline and following GOS supplementation using paired *t*-tests with false-discovery rate (FDR) adjustment for multiple comparisons. Unsupervised principal component analysis (PCA) was used to model the variation in gene expression between baseline and prebiotic intervention. Values of *p* < 0.05 were considered statistically significant. Statistical analyses were performed using SPSS version 24.0 software (Chicago, IL, USA).

## 3. Results

### 3.1. Recruitment and Demographics

From February to October 2017, 19 patients were screened and 18 were recruited. Recruitment and withdrawal reasons are shown in Appendix A. One patient felt unable to comply with the protocol and so withdrew from the study after giving consent but prior to starting the intervention. Seventeen patients returned baseline stool and symptom diaries and were defined as the clinical ITT population. Of the 17 patients in the ITT population, 9 had distal/left sided disease, 5 had proctitis, and 3 had pancolitis. None of the recruited cohort had previously undergone cholecystectomy; therefore, this was not considered as a possible explanation for their gastrointestinal symptoms.

One patient was withdrawn from the trial due to a relapse that required a change to their medication; this was evaluated by the investigators as not being related to the prebiotics. Two patients were withdrawn due to the necessity for antibiotics (one for cystitis, one for tonsillitis) and one patient underwent full bowel preparation for a capsule endoscopy and so was withdrawn for protocol violation. Therefore 13 patients completed the trial and were considered the PP population.

The PP population for peripheral blood microarray analysis of gene expression consisted of 12 patients, as one blood sample had insufficient RNA.

Patient baseline demographics and medication use are described in Appendix A.

One patient failed to return sachets for counting at their final study visit but reported 100% adherence throughout the trial. All 13 patients that completed the trial reported >80% adherence to consuming the sachets over the 6-week period and therefore met the a priori definition of satisfactory adherence. The average adherence measured by counting remaining sachets was 96% (range: 78–100%).

### 3.2. Peripheral Blood Gene Expression

Unsupervised PCA analysis of all expressed genes was used to visually compare the peripheral blood expressed gene (RNA) profiles between baseline and prebiotics in the 12 patients with paired samples in the laboratory PP population (Figure 1).

Seven genes met the 1.5-fold change threshold following GOS supplementation (*p* < 0.05). The five genes upregulated included three coding and two noncoding genes. The two downregulated were both coding genes (Table 1). However, when *p*-values were adjusted for false discovery rate, there were no significant differences in immune gene expression between baseline and prebiotics.

### 3.3. Fecal Outcomes

One participant at baseline and another after prebiotics had calprotectin values below the limit of detection and were assigned values of 0. In the PP (*n* = 13), there was no difference in fecal calprotectin between baseline (585 µg/g) and prebiotics (495 µg/g, *p* = 0.354) (Appendix A). However, there was great variability in responses, with six participants experiencing a reduction of 100 µg/g or more and three with an increase of 100 µg/g or more in fecal calprotectin.

There were no significant differences between baseline and prebiotics for any of the SCFAs or pH in the PP population (*n* = 13) (Appendix A).

For microbiota sequencing, between 1.6 and 2.1 million quality-filtered sequences were obtained for the bacterial 16S rRNA gene. A total of 178 distinct OTUs were assigned a taxonomy at 97% similarity, and data for diversity and relative abundance were compared between baseline and prebiotics in the PP population (*n* = 13).

There were no differences in the Chao1 (50.4 ± 7 vs. 35.6 ± 10, *p* > 0.05) and Shannon (2.5 ± 0.3 vs. 2.2 ± 0.4, *p* > 0.05) diversity indices between baseline and following prebiotic supplementation.

Following prebiotic supplementation, the genus *Oscillospira* and the genus *Dialister* were reduced and the genus *Anaerostipes* was increased (Table 2). Despite evidence that GOS increases bifidobacteria in healthy humans [32], we observed no change in *Bifidobacterium* between baseline (mean 2.23 (SD 3.74)) and prebiotic (3.81 (4.89), *p* = 0.272) when the full dataset was compared.

To explore the reason for the lack of bifidogenesis we performed sub-analyses of patients who at baseline were in remission (SCCAI score of ≤2; despite clinician opinion and objective evidence of active disease) and those who were not (Figure 2). We found that, in those in remission at baseline, the genus *Bifidobacterium* increased from a mean of 1.05% (SD 1.27) to 3.99% (5.37, *p* = 0.046) and the family *Christensenellaceae* increased from a mean of 0.58% (1.27) to 1.25% (2.47, *p* = 0.043) (Table 2), but no other significant changes occurred. In patients not in remission at baseline, there was no change in the genus *Bifidobacterium* between baseline (mean 3.23%, SD 4.91) and prebiotics (3.66%, 4.86, *p* = 0.753), but there was a decrease in the percentage abundance of the genus *Dialister* after 6 weeks (Table 2).

The relationship between baseline bifidobacteria (%) and change in bifidobacteria (%) showed a negative correlation (Pearson’s r2 = −0.319) but this was not significant (*p* = 0.312).

### 3.4. Clinical Outcomes

In the clinical ITT population (*n* = 17), the SCCAI decreased from a mean of 3.3 (SD 2.2) to 2.8 (2.9) following prebiotics; however, this was not statistically significant (*p* = 0.330) (Figure 3a). In the clinical PP population (*n* = 13), the SCCAI fell from a mean of 2.8 (SD 2.1) to 2.4 (3.2), but again this was not statistically significant (*p* = 0.438).

In the clinical ITT population (*n* = 17), there were no significant differences for any of the individual components of the SCCAI, except for a reduction in the severity of stool urgency from a mean of 1.0 (SD 0.7) to 0.6 (0.7, *p* = 0.011) (Figure 3b), between baseline and prebiotic. In the clinical PP population (*n* = 13), the severity of stool urgency decreased from a mean of 1.0 (0.7) to 0.5 (0.8, *p* = 0.020).

There was no difference in the number of patients in clinical remission (SCCAI score ≤ 2) between baseline and prebiotics in either the ITT (7/17 (41%) vs. 11/17 (65%), *p* = 0.219) or the PP (6/13 (46%) vs. 10/13 (77%), *p* = 0.219) populations.

Data for the incidence (out of 7 days) and severity (0 = none, 1 = mild, 2 = moderate, 3 = severe over 7 days) of symptoms reported using the average of the seven-day GSRS diaries are presented in Table 3. There were no significant differences in the incidence or severity of symptoms between baseline and prebiotics except for a reduction in the incidence (mean: 3.2 (SD 2.4) vs. 1.6 (2.0), *p* = 0.012) and severity (mean: 0.7 (SD 0.7) vs. 0.5 (0.8), *p* = 0.046) of loose stool.

Stool form and stool frequency data at baseline and after prebiotic supplementation are presented in Table 3. There was an increase in the proportion of normal stools between baseline (mean 49%, SD 34%) and prebiotics (70%, 36%) (*p* = 0.026).

There were no differences in energy or nutrient intakes between baseline and prebiotics except for alcohol intake, which increased in 6 week diaries compared to baseline (Appendix A).

## 4. Discussion

We report findings from the first trial exploring the effect of GOS prebiotics on peripheral blood immune gene expression and microbiota in UC. It was anticipated that this would allow hypothesis generation regarding the mechanisms of action of GOS in GI inflammation. The data presented here are not strong enough to form a hypothesis about the role of GOS in moderating inflammation. However, the effect of GOS on clinical outcomes is similar to that seen in an RCT of GOS with the low FODMAP diet in IBS [33], and the hypothesis that GOS reduces loose stools and urgency in UC should be tested in a randomized trial. Further, interventions to address this have been highlighted as an important knowledge gap by the National Institute for Health Research.

The hypothesis that supplementation with GOS would modulate peripheral blood markers of immunity and inflammation in UC was based on three previous observations. Firstly, fecal calprotectin was significantly reduced compared to baseline after 2 weeks of prebiotics added to standard medication in UC [11]; secondly, IL-10, IL-8, and natural killer cell activity were upregulated and IL-1β was downregulated compared to placebo in peripheral blood following 1- weeks of GOS supplementation in an elderly population [17]; and thirdly, in vivo studies have demonstrated that short-chain prebiotics modulate immunity via direct interaction with intestinal immune receptors [34]. Our study identified differences in the expression of eight human genes, albeit these were not statistically significant following FDR adjustment. In addition, there was no impact on fecal calprotectin and, in keeping with previous trials of prebiotics in IBD, there was no effect on clinical activity [11,12].

The greatest upregulation of gene expression following GOS supplementation was for phophoglucomutase-5. Clinically relevant phosphoglucomutase deficiency is treated with galactose supplementation, potentially explaining the change [35]. However, in vitro work has demonstrated that increased phophoglucomutase-5 may suppress colon cancer cell progression and migration, alluding to an anti-inflammatory role [36].

CXCL8 was also upregulated and is a noncoding gene isotope of interleukin-8 (IL-8) located within the IL-8 coding region, indicating a possible role in regulation of the gene. Interleukin-8 is a pro-inflammatory cytokine, and bacteria-derived inflammatory signals lead to stimulation of the NF-kB pathway that induces IL-8 via TLR2 stimulation [37]. In elderly adults, IL-8 concentration was increased in peripheral blood mononuclear cells following GOS supplementation [17]. Therefore, ours is the second study to find that GOS may be involved in the IL-8 pathway in peripheral blood.

Three probes for RPL21 and its pseudogene were upregulated following GOS. RPL21 is upregulated in murine platelet cells following *Chlamydia pneumoniae* infection, implicating it in immune response to bacterial infection. In addition, *C. pneumoniae* has been shown to stimulate the TLR2 immune response by innate immune cells [38].

Overall, the increased expression of RPL21 and the CXCL8 noncoding motif suggests that the 6 week intervention may have led to increased expression of genes associated with inflammation rather than immunoregulation. However, the fold changes from baseline were small and, once adjusted for multiple testing, statistically nonsignificant. It is possible that, as has been shown in in vitro studies [16], this difference in expression is a result of immuno-modulatory ‘priming’ of the immune system by the prebiotic, which may have a protective effect against cell damage. Alternatively, it may be that GOS increases an inflammatory response; however, the clinical findings do not support this.

There was no significant effect of the prebiotic on fecal calprotectin or global clinical score (SCCAI); however, some individual symptoms were impacted. Following the prebiotic, lower stool urgency was reported on the SCCAI; however, neither the severity nor the frequency of urgency improved when reported on the GSRS, which, given that it is collected prospectively in a seven-day diary, is likely to be a more accurate reflection of patient experience than the SCCAI for individual symptoms. However, GOS beneficially impacted stool form, with the incidence and severity of loose stool being lower and the number of normal-consistency stools being higher. GOS has previously been shown to reduce the incidence of traveler’s diarrhea [39] and, in IBS, GOS reduced the incidence and severity of loose stool, albeit in combination with the LFD [33]. GOS is a soluble fiber; however, its lack of viscosity and relatively low dose (2.8 g/d) would mean that, mechanistically, these effects are unlikely to be due to increasing stool bulk. An alternative explanation may be that GOS reduces inflammation and inflammatory diarrhea, although the lack of impact on peripheral gene expression and fecal calprotectin refutes this. Finally, effects on reducing inflammation may occur through the modulation of the localized expression of mucous layer genes, thereby reducing urgency and loose stool; however, this theory requires further research in humans [19].

The lack of differences in fecal SCFAs was not anticipated as previous studies of GOS supplementation had reported this [40]. However, background diet was not controlled for in the current study and variation in fermentable carbohydrate intake between baseline and 6 weeks cannot be ruled out as a potential confounder of these findings. Further, the variant level of inflammation in the patient group may affect SCFA production by the microbiota and subsequent absorption by colonocytes, and reduced saccharolytic activity of the microbiota has been reported in IBD [41].

The lack of increase in *Bifidobacterium* following GOS supplementation was unexpected as prebiotics have consistently been shown to increase this genus [42]. However, upon subgroup analysis of patients with less disease activity, an increase in bifidobacteria was identified, indicating that prebiotics may only stimulate bifidogenesis in a less inflamed gut. This finding is consistent with a previous study of ITF prebiotics in active Crohn’s disease that found 4 weeks of ITF supplementation did not increase *Bifidobacterium* [12]. The hypothesis that prebiotics are more effective in a less inflamed gut is further supported by a recent paper that showed healthy siblings of patients with Crohn’s disease, while having similar levels of bifidobacteria at baseline, experienced a greater bifidogenic response after prebiotic supplementation than their siblings that had Crohn’s disease [43].

We chose to use a 6-week treatment period in order that the prebiotic would have time to modulate immune function. In the previously described elderly population study, participants consumed GOS for 10 weeks. However, as the patients recruited into our study had active UC, a longer duration in the current study would have posed a greater risk of patients going into remission due to concurrent medication or relapsing due to the changeable nature of IBD. Previous prebiotic interventions in IBD have ranged from 2 weeks to 12 months [11,44], and so there is no consensus on the length of time that prebiotic interventions are provided for; however, a recent study has suggested that a minimum period of 3 weeks is required for gastrointestinal symptom improvement [45].

As an exploratory study, these data may support a role for GOS in improving loose stool and urgency in UC, although a controlled trial would be required to confirm this. It remains unclear if clinical effects occur via regulation of inflammation, as GOS did not significantly impact immune gene expression in peripheral blood or reduce fecal calprotectin. Future studies should encompass analysis of mucosal gene expression and serum cytokines to further understand how GOS improves symptoms without bacterial and SCFA modulation.

### 4.1. Limitations

There were several limitations that impact on the ability to draw conclusions from the results. The major limitation was the lack of a control group, meaning that differences between baseline and 6 weeks cannot be attributed to the prebiotic supplementation as it is not possible to ascertain if differences were due to natural disease progression, medication, or placebo effect. It would have been feasible to recruit a control group; however, the aim was to give as many patients the active intervention as possible in the time available for recruitment to measure the exploratory outcomes.

The inclusion criteria comprised a subjective assessment of mildly active UC by a gastroenterologist supported by one or more objective measures of disease activity. An alternative approach, which may have been more robust, would have been to base clinical disease activity for inclusion solely on the SCCAI, which may have provided a more homogenous active population group to study. Further, a more sensitive tool could additionally be used to measure clinical improvement, as the mean SCCAI score in our PP population was 2.8 and, therefore, significant improvement from this score may have been difficult to capture.

The sample size was small; however, previous studies investigating the effect of prebiotics in UC have also recruited small numbers. A randomized controlled parallel trial of a 2 week inulin-type fructan intervention in patients with mild to moderate UC recruited 19 patients (10 in the intervention group) [11] and a 4 week randomized controlled study of synbiotics in active UC recruited 18 patients (8 in the intervention group) [46].

### 4.2. Clinical Relevance and Recommendations for Future Research

The findings presented here do not provide evidence to alter clinical practice; however, the prebiotics did not make GI symptoms worse and potentially improved stool form and reduced loose stool and urgency, clinically relevant findings for a patient group that frequently suffers these debilitating symptoms. We set out to observe the relationship between prebiotic supplementation and inflammation in stable active UC. While the effects on gene expression and inflammation were inconclusive, future placebo-controlled research would allow greater understanding of how prebiotics may affect clinical symptoms and alter inflammatory pathways in gastrointestinal inflammation.

## 5. Conclusions

Peripheral blood gene expression showed small fold-changes in seven genes, two of which may be associated with regulation of bacteria-induced inflammation; however, adjustment for multiple testing showed no differences in gene expression. Patients with UC reported an improvement in stool consistency, reduced incidence and severity of loose stools, and less urgency to open their bowels following 2.8 g/d of GOS for 6 weeks. A controlled study investigating the effect on bowel function is essential to determine if GOS prebiotic is a useful adjunct therapy in active UC.

## Figures and Tables

**Figure 1 nutrients-13-03598-f001:**
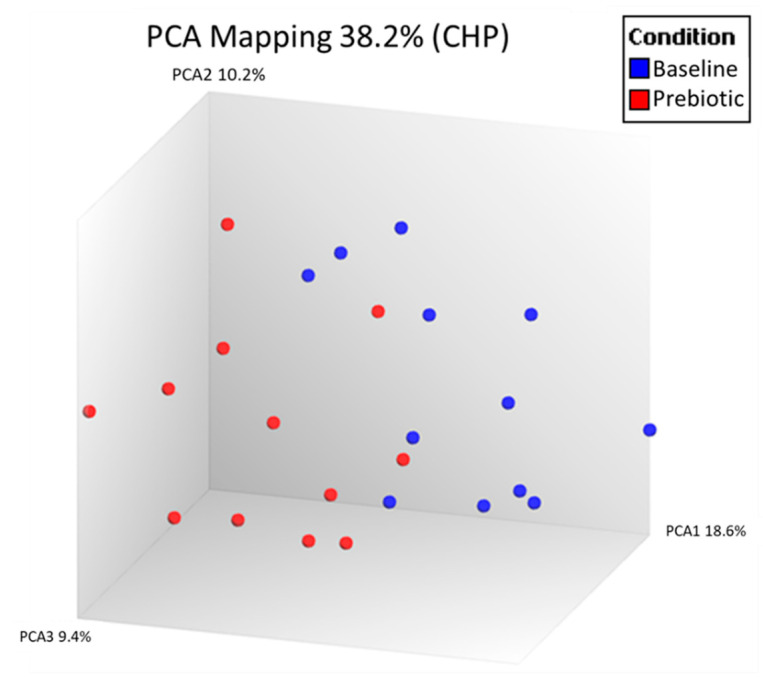
Principal component analysis (PCA) of all expressed genes in the peripheral blood comparing baseline and 6 weeks of prebiotic supplementation in an open-label study of GOS prebiotics in active ulcerative colitis (*n* = 12). Each dot represents one gene expression profile.

**Figure 2 nutrients-13-03598-f002:**
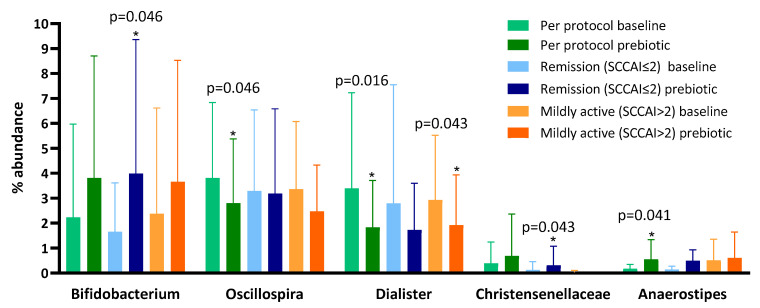
Microbiota proportions compared between baseline and prebiotics (6 weeks) in the per protocol, remission at baseline (Simple Clinical Colitis Activity Score (SCCAI) score ≤ 2), and not in remission at baseline (SCCAI score > 2) populations in a 6-week open-label study of GOS prebiotics in active ulcerative colitis.

**Figure 3 nutrients-13-03598-f003:**
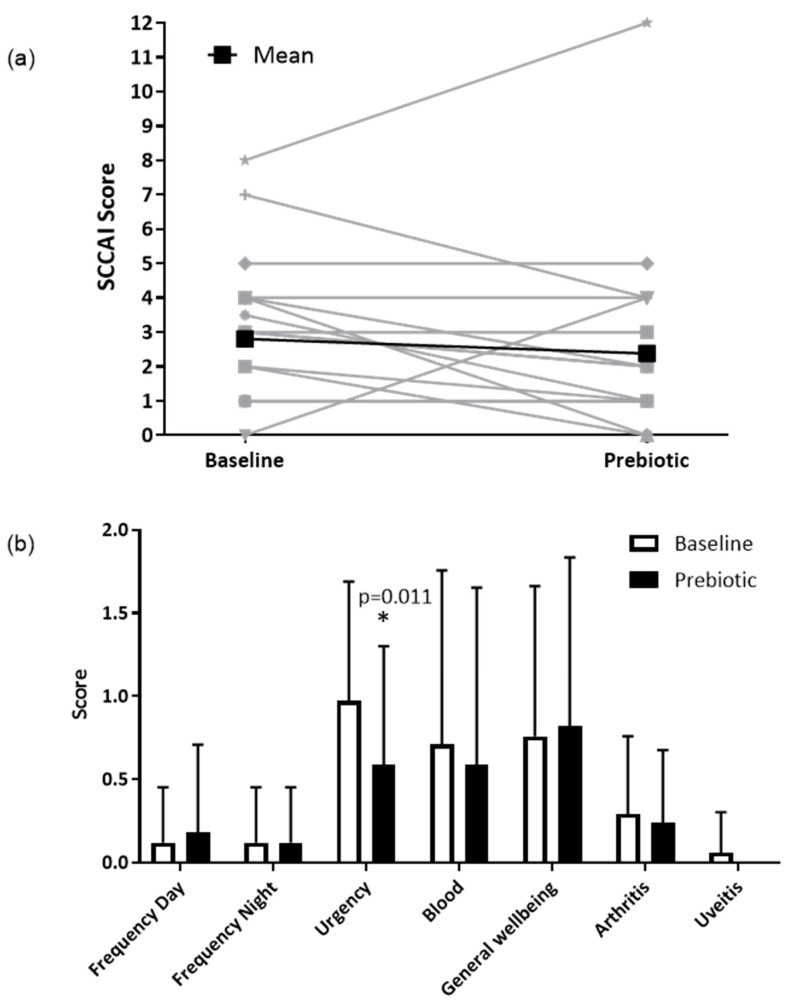
Simple clinical colitis activity index in the ITT population (*n* = 17). (**a**) Total SCCAI score at baseline and after 6-weeks of prebiotic supplementation in a 6-week open-label study of GOS prebiotics in active ulcerative colitis. Gray lines indicate individual patient data, black lines indicate the mean value. (**b**) Simple clinical colitis activity index subscale scores presented as means (SD); a higher score indicates worse symptoms. * *p* = 0.011 (Wilcoxon signed-rank test), uveitis was not present in any participants at 6 weeks.

**Table 1 nutrients-13-03598-t001:** Genes with ≥1.5 log_2_-fold difference in expression between baseline and prebiotics with 6 weeks GOS prebiotics in active ulcerative colitis (*n* = 12).

Public Gene IDs	Gene Symbol	Description	Chr	Strand	Start	Stop	Group	Signal Intensity, Mean (SD) (1)		Log_2_ Fold Change	*p*-Value (2)	FDR *p*-Value (3)
								Baseline	Prebiotic			
**-**	**-**	Phosphoglucomutase (PGM5)	Chr9	+	68328308	68531061	Coding	5.17 (0.53)	6.46 (0.22)	+2.5	0.0007	0.979
AJ227913	CXCL8	Chemokine (C-X-C motif) ligand 8	Chr4	+	73740506	73743716	Noncoding	6.88 (0.39)	7.77 (0.44)	+1.9	0.0068	0.979
NR_031684; uc021uim.1	MIR302F	MicroRNA 302f	Chr18	+	30298910	30298960	Coding	3.82 (0.2)	4.47 (0.26)	+1.6	0.0092	0.979
-	RPL21; RPL21P28	Ribosomal protein L21; ribosomal protein L21 pseudogene 28	Chr13	+	27251309	27256691	Coding	9.87 (0.28)	10.47 (0.26)	+1.5	0.0176	0.979
			Chr1 (4)	+	212051524	212052006	Noncoding					
BC070185	RPL21	Ribosomal protein L21	Chr13	+	27251309	27256691	Coding	16.03 (0.3)	16.64 (0.3)	+1.5	0.0317	0.979
NR_026911	RPL21P28	Ribosomal protein L21 pseudogene 28	Chr1	+	212051524	212052006	Noncoding	14.23 (0.29)	14.82 (0.3)	+1.5	0.0327	0.979
-	-	GRCh38.p12 primary assembly	Chr11	-	45689068	45689092	Coding	7.04 (0.21)	6.43 (0.18)	−1.5	0.0051	0.979
-	-	GRCh38.p12 primary assembly	Chr22	-	23610591	23610615	Coding	6.92 (0.29)	6.29 (0.24)	−1.6	0.0081	0.979

(1) Signal intensity data are expressed as the log_2_ of Tukey’s bi-weight mean (SD), (2) *p*-value obtained via paired *t*-test, (3) FDR *p*-value significance after adjustment for false-discovery rate (Benjamini–Hochberg). (4) For this gene probe both the pseudogene and the coding gene were detected and quantified together. Results were considered statistically significant if *p* < 0.05. Chr, chromosome. FDR, false-discovery rate.

**Table 2 nutrients-13-03598-t002:** Percentage abundances of fecal microbiota compared between baseline and prebiotics in the per protocol, remission at baseline (Simple Clinical Colitis Activity Score (SCCAI) score ≤ 2), and not in remission at baseline (SCCAI score > 2) populations in a 6-week open-label study of GOS prebiotics in active ulcerative colitis. The taxa with the top 10 Z-scores are presented for each group comparison.

Mean (SD)						
Per Protocol Population (*n* = 13)	Baseline		Prebiotic		Z-Score	*p*-Value *
p__Firmicutes_g__*Dialister*	3.39	(3.84)	1.83	(1.88)	−2.411 b	**0.016**
p__Firmicutes_g__*Anaerostipes*	0.17	(0.17)	0.55	(0.79)	−2.040 c	**0.041**
p__Firmicutes_g__*Oscillospira*	3.81	(3.03)	2.80	(2.58)	−1.992 b	**0.046**
p__Firmicutes_f_[*Mogibacteriaceae*]	0.38	(0.43)	0.24	(0.24)	−1.883 b	0.06
p__Firmicutes_g__*Dorea*	1.24	(0.79)	1.56	(0.91)	−1.852 c	0.064
p__Bacteroidetes_g__*Paraprevotella*	0.04	(0.08)	0.14	(0.27)	−1.826 c	0.068
p__Proteobacteria_g__*Haemophilus*	2.86	(9.53)	0.26	(0.74)	−1.782 b	0.075
p__Firmicutes_f__*Erysipelotrichaceae*	0.86	(1.54)	1.58	(3.20)	−1.642 c	0.101
p__Actinobacteria_g__*Corynebacterium*	0.00	(0.00)	0.01	(0.04)	−1.604 c	0.109
p__Firmicutes_g__*Anaerotruncus*	0.03	(0.05)	0.02	(0.04)	−1.599 b	0.11
**Patients in remission at baseline (SCCAI≤2) (*n* = 6)**						
p__Firmicutes_ f__*Christensenellaceae*	0.58	(1.27)	1.25	(2.47)	−2.023 c	**0.043**
p__Actinobacteria_g__*Bifidobacterium*	1.05	(1.27)	3.99	(5.37)	−1.992 c	**0.046**
p__Firmicutes_g__*Anaerostipes*	0.17	(0.14)	0.49	(0.44)	−1.782 c	0.075
p__Firmicutes_f__[*Mogibacteriaceae*]	0.24	(0.30)	0.18	(0.29)	−1.753 b	0.08
p__Bacteroidetes_g__*Prevotella*	2.46	(3.81)	0.43	(0.93)	−1.604 b	0.109
p__Firmicutes_g__*Holdemania*	0.03	(0.04)	0.01	(0.02)	−1.604 b	0.109
p__Firmicutes_f__*Ruminococcaceae*	3.66	(4.56)	6.88	(7.66)	−1.572 c	0.116
p__Firmicutes_g__*Dorea*	0.93	(0.62)	1.49	(0.88)	−1.572 c	0.116
p__Firmicutes_g__*Dialister*	3.25	(5.04)	1.72	(1.88)	−1.363 b	0.173
p__Actinobacteria_g__*Slackia*	0.03	(0.08)	0.05	(0.08)	−1.342 c	0.18
**Patients not in remission at baseline (SCCAI>2) (*n* = 7)**						
p__Firmicutes_g__*Dialister*	3.50	(2.87)	1.92	(2.02)	−2.028 b	**0.043**
p__Firmicutes_g__*Roseburia*	3.27	(2.35)	5.86	(3.78)	−1.859 c	0.063
p__Firmicutes_g__*Blautia*	2.74	(1.83)	4.60	(3.38)	−1.859 c	0.063
p__Proteobacteria_g__*Sutterella*	0.22	(0.20)	0.41	(0.28)	−1.859 c	0.063
p__Firmicutes_g__*Oxobacter*	0.06	(0.08)	0.02	(0.04)	−1.826 b	0.068
p__Cyanobacteria_o__Streptophyta	0.03	(0.04)	0.01	(0.03)	−1.753 b	0.08
p__Proteobacteria_g__*Bilophila*	0.07	(0.08)	0.22	(0.22)	−1.753 c	0.08
p__Firmicutes_g__*Oscillospira*	3.81	(3.11)	2.47	(1.86)	−1.690 b	0.091
p__Firmicutes_g__*Anaerofilum*	0.01	(0.01)	0.00	(0.01)	−1.604 b	0.109
p__Proteobacteria_g__*Haemophilus*	5.03	(13.02)	0.41	(1.02)	−1.521 b	0.128

* *p* values for symptoms are the result of a Wilcoxon signed-rank test, *p* was considered statistically significant if <0.05 and values reaching this threshold are marked in bold. *p* = phylum; o = order; f = family; g = genus; b = based on positive ranks; c = based on negative ranks.

**Table 3 nutrients-13-03598-t003:** Incidence and severity (0–3) of gastrointestinal symptoms reported on the seven-day Gastrointestinal Symptom Rating Scale and stool frequency and form reported on the Bristol Stool Form Scale at baseline and after 6 weeks of GOS prebiotics in active ulcerative colitis.

	Incidence (Days Out of 7) (*n* = 17)					Severity (^ = 17)				
Mean (SD)	Baseline		Prebiotic		*p* *	Baseline		Prebiotic		*p* *
Pain	2.9	(2.5)	3.1	(2.9)	0.650	0.6	(0.6)	0.6	(0.7)	0.592
Heartburn	0.5	(1.2)	0.6	(1.3)	0.854	0.1	(0.3)	0.1	(0.2)	0.854
Acid reflux	0.3	(0.8)	0.1	(0.3)	0.257	0.1	(0.2)	0.0	(0.1)	0.197
Nausea	0.9	(1.0)	1.4	(2.1)	0.478	0.2	(0.2)	0.2	(0.4)	0.776
Gurgling	4.0	(2.8)	3.3	(2.7)	0.084	0.7	(0.6)	0.6	(0.6)	0.105
Bloating	2.8	(2.7)	3.3	(2.9)	0.361	0.6	(0.6)	0.6	(0.6)	0.443
Belching	2.7	(3.0)	3.1	(3.0)	0.429	0.4	(0.5)	0.4	(0.4)	0.794
Flatulence	4.2	(2.8)	3.8	(3.0)	0.228	0.9	(0.6)	0.7	(0.6)	0.089
Constipation	1.1	(1.9)	1.1	(1.6)	1.000	0.2	(0.4)	0.2	(0.3)	0.917
Diarrhea	1.5	(2.8)	0.9	(2.1)	0.180	0.4	(0.8)	0.3	(0.8)	0.285
Loose stool	3.2	(2.4)	1.6	(2.0)	**0.012**	0.7	(0.7)	0.5	(0.8)	**0.046**
Hard stool	0.5	(1.0)	0.1	(0.2)	0.066	0.1	(0.3)	0.0	(0.1)	0.068
Urgency	2.8	(2.4)	2.4	(2.5)	0.319	0.6	(0.6)	0.5	(0.7)	0.277
Incomplete evacuation	1.6	(2.1)	1.3	(2.0)	0.339	0.3	(0.5)	0.2	(0.3)	0.182
Tiredness	3.5	(2.9)	3.3	(3.0)	0.760	0.8	(0.8)	0.8	(0.9)	0.681
Overall symptoms	3.8	(3.0)	3.8	(2.8)	0.609	0.7	(0.7)	0.8	(0.7)	0.964
**Stool output**										
Frequency, /d, mean (SEM)	2.0	(1.2)	2.1	(1.4)	0.833					
Form, % of stools, mean (SEM)										
Hard stool	13%	(29)	1%	(3)	0.117					
Normal stool	49%	(34)	70%	(36)	**0.026**					
Soft stool	39%	(35)	29%	(36)	0.132					

Values are presented as means (standard deviation). * *p* values for symptoms are the result of a Wilcoxon signed-rank test and for stool output they are the result of paired *t*-tests. *p* was considered statistically significant if <0.05 and values reaching this threshold are marked in bold.

## Data Availability

The data presented in this study are available on request from the corresponding author. The data are not publicly available due to limitations provided in the ethical approval of the study.

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
