# Peer review of "Prebiotic Galactooligosaccharide Supplementation in Adults with Ulcerative Colitis: Exploring the Impact on Peripheral Blood Gene Expression, Gut Microbiota, and Clinical Symptoms"

_nutrients, 2021, doi:10.3390/nu13103598_

Round 1

Reviewer 1 Report

The manuscript has a lot of limitations, which are acknowledged by the authors and therefore are the results only with confined interpretability/clinical usefulness. Nevertheless, the work of the authors is very well presented and might inspire further large-scale studies in the same or similar context.

Below you will find several proposals to help further improve your study

  1. Latin/Greek scientific terms in italics e.g. in vivo, in vitro,Oscillospira etc (as well as gene names)
  2. Calprotectin 150 μg cut-off. Please cite relevant validating study
  3. Chologenic diarrhea post colocystectomy may influence the clinical semiology of UC patients. Please explain why such patients were included
  4. Please give your arguments for choosing SCCAI for scoring ulcerating colitis and cite, respectively (since you implicate gastroenterologists and endoscopic localization, but not endoscopic score..)
  5. Please provide ethical approval number and state whether study complies with GCP and Helsinki principles

Author Response

Thank you so much for your feedback on our manuscript Prebiotic galacto-oligosaccharide supplementation in adults with ulcerative colitis: exploring the impact on peripheral blood gene expression, gut microbiota, and clinical symptoms. We are delighted that you found our research to be well presented, interesting and important. We agree that there are limitations to the study but hope that this publication will help to guide further research in this area.

Response to reviewer 1 comments:

  1. Latin/Greek scientific terms in italics e.g. in vivo, in vitro,Oscillospira etc (as well as gene names)

Thank you so much for your review of our manuscript, apologies for the oversight in the italicising, this occurred when copying the manuscript into the ‘Nutrients’ document format following which we seem to have lost some of our formatting. We have addressed this now.

  1. Calprotectin 150 μg cut-off. Please cite relevant validating study.

Thank you so much for this observation. The value of 150ug/g was selected  as studies have identified a calprotectin of >150mg/g as correlating with active disease. Gisbert JP,  et al. Inflamm Bowel Dis. 2009 Aug;15(8):1190-8.

This has now been added to the text in the methods.

  1. Chologenic diarrhea post cholecystectomy may influence the clinical semiology of UC patients. Please explain why such patients were included.

Thank you for this comment. We understand that this can occur post cholecystectomy but didn’t think this would be an over-looked major causative factor of diarrhoea in this patient group. However, importantly, no patients in the recruited cohort had undergone this procedure and therefore will have diarrhoea of this aetiology. This has now been added to the results section (line 248-250).

  1. Please give your arguments for choosing SCCAI for scoring ulcerating colitis and cite, respectively (since you implicate gastroenterologists and endoscopic localization, but not endoscopic score..).

We required objective evidence of active disease as an inclusion criteria.  However, SCCAI was chosen as a disease activity indicator and outcome measure as it has proven validity compared with more complex disease activity scoring indices  and  allows assessment over time without subjecting the participant to further laboratory testing or invasive procedures.  We have now added this sentence to the methods sections to clarify the reason for our choice of clinical scoring tool.

  1. Please provide ethical approval number and state whether study complies with GCP and Helsinki principles

Thank you for your feedback on this. When formatting the paper for submission using the standard template for “Nutrients’ it seemed the correct place to include this was at the end of the paper under the Institutional Review Board Statement: The study was conducted according to the guidelines of the Declaration of Helsinki and approved by the Yorkshire and The Humber- Bradford Leeds NHS Research Ethics Committee (16/YH/0438) and the Health Research Authority. We have therefore left the statement there in line with the journal guidelines and template.

Reviewer 2 Report

The subject reviewed by researchers is both interesting and important, especially because the exact cause of UC is unknown. The authors described the methods and results in a clear informative way. Also, the figures and tables provided by the authors are very clear and informative. 

In discussion, the authors marked that their data is not sufficient to form a hypothesis about the role of GOS in moderating inflammation and as they mentioned later the changes they observed might serve as a base to perform a controlled study. The limitations are well described. 

- provide more information about IBD itself (and/or studied UC) in Introduction

- check punctuation mistakes and abbreviations e.g. CD

Author Response

Thank you so much for your feedback on our manuscript Prebiotic galacto-oligosaccharide supplementation in adults with ulcerative colitis: exploring the impact on peripheral blood gene expression, gut microbiota, and clinical symptoms. We are delighted that you found our research to be well presented, interesting and important. We agree that there are limitations to the study but hope that this publication will help to guide further research in this area.

Response to reviewer 2 comments:

- provide more information about IBD itself (and/or studied UC) in Introduction

Many thanks for this advice on how we could improve our manuscript, we agree that by adding a paragraph on IBD to the introduction it strengthens the manuscript. We have now done this (line 50-56)

- check punctuation mistakes and abbreviations e.g. CD

Many thanks for your observations in this we have decided not to abbreviate Crohn’s disease, as this disorder is not the focus of the manuscript and we have tried to keep its mention to a minimum. We  have performed a full check of punctuation throughout the manuscript, we hope that you will find this acceptable.